# Stereotactic Radiotherapy for Oligometastasis

**DOI:** 10.3390/cancers11020133

**Published:** 2019-01-23

**Authors:** Sotaro Otake, Taichiro Goto

**Affiliations:** Lung Cancer and Respiratory Disease Center, Yamanashi Central Hospital, Kofu 400-8506, Japan; sotaro.otake@gmail.com

**Keywords:** cancer, oligometastasis, surgery, stereotactic ablative body radiotherapy, oncology

## Abstract

Oligometastatic disease is defined as “a condition with a few metastases arising from tumors that have not acquired a potential for widespread metastases.” Its behavior suggests a transitional malignant state somewhere between localized and metastatic cancer. Treatment of oligometastatic disease is expected to achieve long-term local control and to improve survival. Historically, patients with oligometastases have often undergone surgical resection since it was anecdotally believed that surgical resection could result in progression-free or overall survival benefits. To date, no prospective randomized trials have demonstrated surgery-related survival benefits. Short courses of highly focused, extremely high-dose radiotherapies (e.g., stereotactic radiosurgery and stereotactic ablative body radiotherapy (SABR)) have frequently been used as alternatives to surgery for treatment of oligometastasis. A randomized study has demonstrated the overall survival benefits of stereotactic radiosurgery for solitary brain metastasis. Following the success of stereotactic radiosurgery, SABR has been widely accepted for treating extracranial metastases, considering its efficacy and minimum invasiveness. In this review, we discuss the history of and rationale for the local treatment of oligometastases and probe into the implementation of SABR for oligometastatic disease.

## 1. Introduction

Cancer is widely regarded as a systemic disease. Historically, lung cancer with distant metastases was regarded as stage IV or end-stage disease and indicated for palliative treatment. Pharmacotherapy has been the standard treatment choice for metastatic cancer, including cytotoxic chemotherapy, molecular-targeted therapy, and hormonal manipulation. Notwithstanding recent advances in pharmacotherapy, prognosis of metastatic cancer remains unsatisfactory [1,2,3]. Within the metastatic cancer patient population, it is becoming apparent that a fraction of patients have an oligometastatic state; in this state, local therapy for metastatic lesions results in satisfactory survival that is comparable to that in non-metastatic disease. The concept that patients with only a limited number of metastases from a malignant tumor can potentially be cured was developed in 1995 and was termed “oligometastasis”, describing an intermediate stage between localized and metastasized cancer [4]. There is currently no consensus as to the definition of oligometastatic disease; however, most clinical trial protocols and clinicians accept a definition of 1–3 or 1–5 metastatic lesions [5,6]. The schema of oligometastases is shown in Figure 1. In oligometastasis, both the primary tumor and distant metastases are considered active but curable.

Based on this concept, medical doctors have started treating these patients using local therapies with curative intents. For instance, resection of liver metastases of colorectal cancer results in long-term survival of patients [7]. Similarly, for non-small-cell lung cancer (NSCLC), it was demonstrated that long-term survival may be achieved for a subset of patients with oligometastasis using local treatment [8]. However, details of treatment vary widely and only a small number of studies have prospectively investigated the optimal treatments for such patients [8,9,10,11,12,13,14,15,16,17,18].

Over the several decades, stereotactic ablative radiotherapy (SABR)—also termed stereotactic body radiation therapy (SBRT) or stereotactic radiosurgery (SRS)—has emerged as a noninvasive alternative to surgery [19,20,21,22]. These technologies have been rapidly adopted in the absence of supporting clinical evidence. Many fundamental questions regarding the oligometastatic state remain unsolved. Only limited randomized data are available that support the hypothesis that ablative treatments improve overall survival (OS) in these patients; the biological characteristics and biomarkers that indicate oligometastatic state also remain unknown. This review aims to elucidate the unknown aspects of oligometastasis and to discuss the emerging role of stereotactic ablative body radiotherapy (SABR) in the management of oligometastatic disease.

## 2. Biology of Oligometastasis

Metastasis occurs when migrating cancer cells colonize a tissue microenvironment other than the primary tumor. Due to the advances in genomic research, evidence has emerged that subclones capable of adapting to the metastatic site microenvironment appear during tumor evolution and these subclones metastasize to the distant organ. Genomic analysis has demonstrated that lung cancer and pulmonary metastasis harbor common driver mutations [23,24]. These findings indicate that metastatic cells emerge from truncal expansion rather than from a completely different subpopulation.

Primary tumors are composed of genetically heterogeneous cell populations [25,26,27,28], and it is presumed that tumor clones do not have equal abilities for metastasization (Figure 2). Using a cell line derived from B16F1 melanoma, Fidler et al. found that cells with variant metastatic potentials pre-exist in a heterogeneous primary tumor [29]. High and low metastatic potentials have also been reported in various cell lines, such as MHCC97 hepatocellular carcinoma cells, PC-14 human lung adenocarcinoma cells, and KHT sarcoma cells [30,31,32]. Metastatic potential is also assumed to be subject to intratumor heterogeneity (variation in an individual cell’s metastatic potential). When cells with low metastatic potential incidentally metastasize, they may induce oligometastasis (Figure 2).

Novel insights into the existence of oligometastasis were advocated by Yachida et al.; by using a next-generation sequencer, they analyzed the genomes of seven patients with metastases from pancreatic cancer to evaluate the clonal relationships between primary and metastatic cancers [33]. A quantitative analysis of the time required for genetic evolution of the pancreatic cancer indicated that at least a decade had passed from occurrence of the initial mutation to birth of the parental, non-metastatic founder cell [33,34]. At least five more years are necessary for acquisition of metastatic potential [33,34]. This broad time window emphasizes that oligometastatic clones might develop prior to polymetastatic clones during tumorigenesis.

Molecular mechanisms that cause oligometastasis have also been investigated from other perspectives. To distinguish oligometastasis from polymetastasis, Wuttig et al. identified genes that characterized few (≤8) or many (≥16) pulmonary metastases using samples from renal cell carcinoma patients [35]. They identified 135 differentially expressed genes by comparing between patients with many (≥16) and few metastases (≤8). Furthermore, polymetastatic tumors showed elevated expression of cell division- and cell cycle-associated genes [35]. These data provide molecular-level evidence for existence of an oligometastatic state, but further study is required to clarify mechanisms that contribute to the development of oligometastatic phenotypes.

## 3. Clinical Evidence of Oligometastasis

Limited evidence is available that supports the hypothesis that local treatment for oligometastasis results in patient benefits. Clear evidence for survival benefits of locally ablative treatment exists for limited brain metastases. In 2004, Andrews et al. published the results of the phase III randomized controlled trial (RTOG9508): The addition of an SRS boost to whole-brain radiotherapy improved median OS from 4.9 to 6.5 months and provided a 1.6 month survival benefit in patients with a single unresected brain metastasis [9]. Notably, improved OS was only found by subgroup analysis, as patients with 2 or 3 brain metastases did not reveal survival benefits [9].

For hepatic metastases of colorectal cancer, a randomized phase II study (CLOCC-trial) examined OS in 119 patients without extrahepatic disease who received either systemic treatment alone or systemic treatment plus local radiofrequency ablation [36]. In 2017, the published long-term follow-up data showed a statistically significant OS benefit (hazard ratio (HR), 0.58; *p* = 0.01) in patients included in the intervention arm at almost 10 years of follow-up. The OS rates at 8 years in the combined therapy arm and systemic treatment arm were 35.9% vs. 8.9%, respectively [37].

Gomez et al. conducted a phase II RCT comparing local consolidative therapy in synchronous oligometastatic NSCLC with three or fewer metastatic lesions [13]. All patients received at least four cycles of platinum-doublet chemotherapy or at least targeted epidermal growth factor receptor (EGFR) or anaplastic lymphoma kinase (ALK) inhibitors for 3 months without progression prior to randomization; then, these patients were randomized to receive either maintenance therapy alone or to receive local consolidative therapy, including SABR, surgery, and/or conventional (chemo)radiotherapy to all known disease sites, with or without maintenance therapy. The study was terminated early due to a significant improvement in the primary endpoint of progression-free survival (PFS) in those undergoing consolidative therapy (median PFS, 11.9 vs. 3.9 months; *p* < 0.05). Among local consolidative therapies, radiotherapy was the prevailing treatment regimen: 96% of patients randomized in the intervention arm received some form of radiotherapy whereas 48% received SABR. Notably, no grade 4 or 5 toxicities were reported [13].

Overall, evidence of benefits from locally ablative treatments—especially SABR—in patients with extracranial oligometastasis is evolving; however, randomized evidence is yet to be established. Evidence-based recommendations for patient selection and optimal combinations of local and systemic treatments are awaited.

## 4. Clinical Implication of Oligometastasis

Patients are increasingly being diagnosed with oligometastatic disease due to the advent of sensitive imaging technologies and effective therapies which allow patients to live longer with cancer diagnoses [38,39]. In NSCLC, 50% of patients who are newly diagnosed with stage IV NSCLC are found to have 3 or less metastases [40]. In case of postoperative recurrent NSCLC with distant metastases, 33% of patients are found to have isolated metastases and 19% are found to have 2–3 metastases [40]. In prostate cancer, 41% of patients with recurrence after local therapy are found to have 5 or less metastases [41]. In breast cancer, 43–77% of patients are found to have 2 or less metastatic lesions during follow-up after systemic chemotherapy [42,43,44,45,46,47]. In previously untreated metastatic colorectal cancer, 38% of patients are found to have isolated metastases and 55–85% are found to have 2–3 localized metastases [48,49,50].

Outcomes of oligometastasis treatments have greatly improved with recent medical advances. Prognosis has originally been demonstrated to be better in patients with oligometastasis than in those with polymetastasis in many types of cancers, such as prostate, breast, and lung cancers [41]. Moreover, it has been reported that aggressive resection of metastatic lesions (in the lung, liver, adrenal gland, and brain) can achieve long-term survival in selected patients [51,52,53]. Prolonged survival prognoses have been reported in patients undergoing metastasectomy compared with those not undergoing the procedure (breast cancer, colorectal cancer, and melanoma) [54,55,56,57]. However, it cannot be ignored that a selection bias or lead-time bias may have influenced these findings. The contribution of local therapy to prolonged survival prognosis in patients with oligometastases greatly varies depending on the cancer type. For lung, breast, and other cancers, many issues still remain unsolved regarding the significance of local therapy whereas for hepatic and pulmonary metastases of colorectal cancer, pulmonary metastasis of osteosarcoma, etc., resection of metastatic lesions is regarded as the standard treatment [58,59].

In patients with hepatic metastasis of colorectal cancer, OS after radical resection is reported to be 25–50% at 5 years and 22% at 10 years [7,60,61]. Furthermore, Tomlison et al. reported that disease-specific death occurred in only one of 102 patients who survived 10 years or more after surgery and that complete cure of cancer was achieved in almost all patients [62]. According to a review of sarcoma-related studies, pulmonary metastasis was observed in 20–40% of patients, but many of them did not present with metastasis to other organs [63]. In sarcoma, lesions that are detectable on imaging are difficult to control with systemic chemotherapy. Thus, resection of pulmonary metastasis plays an important role. Based on retrospective studies, 5-year OS after resection of pulmonary metastasis is reported to be 21–38% [64,65,66,67,68]. Local therapy for these metastatic lesions is also recommended by the National Comprehensive Cancer Network.

## 5. Optimal Selection of Patients with Oligometastatic Disease

Although the improvement of survival with ablative treatments is yet to be validated, several studies have elucidated clinical prognostic factors for long-term survival after ablative therapy. The identified prognostic factors tend to include four major overarching criteria: young age, patient fitness, slow-growing disease (i.e., metachronous metastases or a long disease-free interval (DFI)), and low disease burden (i.e., a smaller number of metastases, small size) [69].

A systematic review of 15 studies by Spelt et al. demonstrated that the number of metastases was a prognostic factor in all prospective studies [70]. Kanzaki et al. studied patients who had undergone pulmonary metastasectomy for metastases from renal cell carcinoma and demonstrated that a DFI greater than or equal to 2 years was associated with a 5-year survival of 58%, but a DFI less than 2 years was associated with a 5-year survival of 26% [71]. In a recent meta-analysis, slow-growing disease and low disease burden were found to have the most prognostic value in patients with oligometastatic NSCLC who were treated with surgery or radiotherapy: low-risk group, patients with metachronous metastases; intermediate-risk group, patients with synchronous metastases and N0 disease; and high-risk group, patients with synchronous metastases and N1 or N2 disease [72]. As to the size of metastasis, several studies identified large size as a poor prognostic factor and included it in their risk score systems [7,61,73,74,75,76,77].

In addition to these four factors, other factors, including tumor histology, have sometimes been shown to play a role in patient prognosis. An early surgical series demonstrated that chemosensitive histologies (such as germ cell tumors) are associated with better OS after resection; this remains an important prognostic factor [78]. Hepatic metastases from neuroendocrine tumors, pulmonary metastases from germ cell tumors, and breast cancer oligometastases are associated with relatively prolonged survival whereas hepatic metastasis from gastric cancer is associated with a worse prognosis. Casiraghi et al. reported that tumor origin predicts prognosis after pulmonary metastasectomy, with 5-year survival rates of 46%, 37%, 39%, and 90% for epithelial cancer, melanoma, sarcoma, and germ cell cancer, respectively [79]. Given the rapidly changing landscape in systemic therapies, with new options available for previously chemoresistant histologies such as melanoma and NSCLC, the role of histology in prognosis may change in the future [80].

So far, only clinical factors are available for patient selection in the clinical setting, and unfortunately, the selection is becoming more loosely defined. Furthermore, among the available risk scores, none of them could predict disease-free survival with sufficient discriminatory accuracy [81]. Thus, limitations in accuracy of the risk scores have made it difficult to utilize them in clinical settings. Rather than relying on clinical factors, identifying and validating biological characteristics that are predictive of oligometastatic disease could enhance the existing clinical criteria for patient selection. Additional biological data may also help patient stratification [82,83].

Expression levels of microRNAs, small non-coding RNAs known to regulate tumor apoptosis and proliferation, are frequently abnormal in cancer and metastasis [84,85,86]. Emerging evidence indicates that microRNA profiles could be useful for distinguishing oligometastasis from polymetastasis. Recently, Lussiter et al., using a xenograft model, found that microRNA-200c expression was associated with polymetastatic progression in an oligometastatic cell line derived from patients treated with high-dose radiotherapy [87]. Uppal et al. identified overexpression of three microRNAs in clinical metastasis samples from patients with oligometastatic disease. MicroRNA-655–3p, microRNA-544a, and microRNA-127–5p were shown to limit metastasis in a model of lung colonization of breast cancer [88]. Further studies and clinical cohorts are essential to validate microRNA profiles of oligometastasis compared to those of polymetastasis.

## 6. Role of SABR in the Treatment of Oligometastasis

SABR is a specialized form of radiation treatment and is characterized by high doses of radiation per fraction (5–34 Gy), shorter time period (few fractions), and an accurate tumor targeting system [69,89,90,91]. SABR has become widely available recently due to technological advancements in both imaging of target tumors and in precise radiation delivery.

Earlier, one of the first machines used to deliver SABR was the Gamma Knife, also termed as SRS, for intracranial tumors [19]. Later, in the mid-1990s, SABR for extracranial targets was accomplished through incremental advances in linear accelerator-based solutions [21]. SABR delivery without a body frame is often made possible through the use of advanced image-guidance (e.g., integrated cone-beam computed tomography [CT)) that allows for correction of alignment using landmarks just before treatment delivery [92]. Additionally, motion management strategies, such as four-dimensional CT imaging, have enabled quantification of tumor motion along the respiratory cycle [93].

Regarding the fractionation used in SABR, there is neither an evidence-based recommended dose nor fractionation at present. However, the ongoing NRG-BR001 trial (a phase 1 study of SABR for treatment of multiple metastases) is currently investigating the recommended SABR dose for each metastatic location when multiple metastases are treated with SABR. Patients with metastatic breast cancer, NSCLC, and prostate cancer are eligible for this study. The allowed metastatic sites are as follows: peripheral lung, central lung, mediastinal/cervical lymph nodes, liver, spine/paraspine, osseous, and abdomen/pelvis. NRG-BR001 outlines a location-adapted approach for multi-organ site ablative radiation therapy (MOSART) SABR. The findings of this study will help in guiding the development of appropriate dosages for different metastatic sites.

To date, no RCTs have been conducted that compare between SABR and surgery for oligometastasis treatment. Historically, the role of radiotherapy in oligometastasis treatment was limited because of the lack of effective systemic therapy and inferiority of radiotherapeutic techniques compared to current standards. However, at present, high doses of radiation can be safely delivered to a few small identified metastatic lesions. Many patients with oligometastasis receive systemic therapy and their physical strength is often impaired. In many, long-term survival is not always achievable; therefore, less invasive local therapy may be desirable in terms of allowing them to live their remaining lives with a better quality of life. SABR appears to provide a high level of local control with minimal associated toxicity [94]; it possibly delays overall progression of disease, and it may influence overall survival. This procedure is highly expected to be used widely as local therapy for oligometastasis in the future.

## 7. Ongoing Clinical Trials

Several ongoing studies are enrolling patients to assess the use of SABR for oligometastases (Table 1) [95]. The UK CORE trial is a multi-institutional randomized phase II study including patients with metachronous oligometastatic NSCLC, breast cancer, and prostate cancer that is randomized to standard of care with or without SABR. Furthermore, in the UK, the Stereotactic Ablative Radiotherapy for Oligometastatic Non-Small Cell Lung Cancer (SARON) trial is evaluating the use of systemic chemotherapy with or without radial radiotherapy for primary disease and up to 3 metastatic sites (NCT02417662). Stereotactic ablative radiotherapy for comprehensive treatment of oligometastatic tumors (SABR-COMET) is an international randomized phase II trial that is enrolling patients with up to 5 metastases (NCT01446744). All patients will be treated with standard of care chemotherapy and randomization with SABR to all known oligometastases or no SABR; the primary endpoint is designed to detect a difference in OS [96]. The Surveillance or metastasis-directed Therapy for OligoMetastatic Prostate cancer recurrence (STOMP) trial on prostate cancer is currently ongoing. With a primary endpoint of androgen deprivation therapy-free survival, the investigators are randomizing patients with metastatic disease to receive either local therapy (surgery or radiation) or active clinical surveillance (NCT01558427). NRG-BR002 is a phase II/ III trial comparing standard of care treatment with standard of care plus SABR in women with 1–2 breast cancer metastases (NCT02364557). The trial is powered to address progression-free survival in the phase II study, and the study will automatically expand to a phase III design if a benefit in progression-free survival is observed in the phase II component. NRG-LU002 is a multicenter randomized phase II/III trial comparing chemotherapy with chemotherapy plus SABR for oligometastatic (1 to 3 lesions) NSCLC. These studies will continue to establish collective data for the utilization of SABR for oligometastases.

## 8. Future Perspectives

Here we reviewed the emerging role of SABR in management of oligometastatic disease; however, several clinical questions remain unsolved, such as how to identify suitable patients for radical local treatment and whether or not SABR should replace surgery. Developing novel biomarkers for predicting treatment response and carefully designed exploratory and confirmatory studies of potential biomarkers are urgent needs for applying this new treatment strategy in the clinical setting. Randomized evidence comparing SABR and surgery is also necessary to guide a treatment option of the oligometastatic disease.

## 9. Conclusions

Considering the abundant clinical and biological evidence that supports the notion of oligometastasis, we infer that oligometastasis is not a hypothetical disease concept but a real therapeutic target for which SABR can achieve high rates of local control. If the pathology of oligometastasis is understood further, we can establish improved biomarkers for accurate identification of patients with oligometastases; thus, selection of locally aggressive therapies may be possible. Moreover, prospective and randomized trials evaluating the efficacy and validity of SABR will also be needed for guiding the treatment of the oligometastatic state.

## Figures and Tables

**Figure 1 cancers-11-00133-f001:**
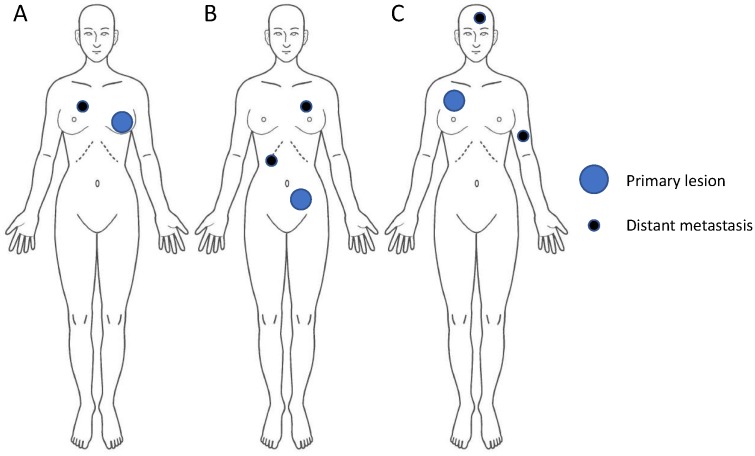
Schema of oligometastasis. Cases **A**, **B,** and **C** represent breast cancer with solitary pulmonary metastasis, colon cancer with liver and lung metastases, and non-small-cell lung cancer with brain and bone metastases, respectively. In oligometastatic disease, the number of metastatic lesions is limited, and both the primary and metastatic lesions should be treated with local treatment.

**Figure 2 cancers-11-00133-f002:**
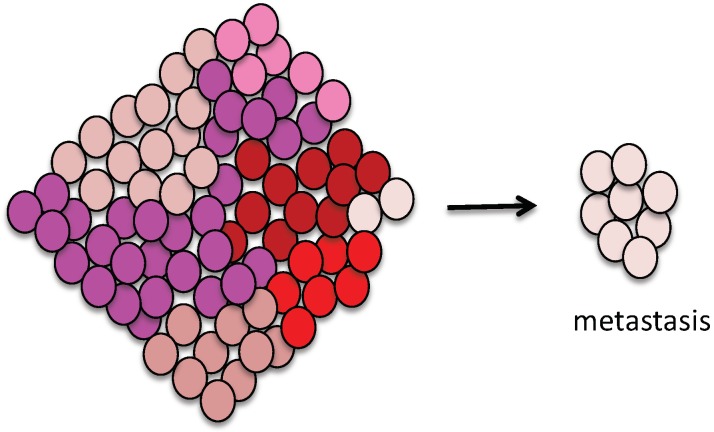
Schema of intratumor heterogeneity regarding metastatic potential. Tumors are composed of cell populations with various metastatic potentials. Denser red colors indicate higher metastatic potential in this schema. In the metastatic phase during cancer evolution, many different selection pressures are generated, and subclones that can adapt to the microenvironment at metastatic sites appear and metastasize. Thus, metastasizing subclones may not have the highest metastatic potential, but possibly the best adaptability to the microenvironment. When cells with low metastatic potential happen to metastasize, it may lead to oligometastatic state.

**Table 1 cancers-11-00133-t001:** Selected open trials of SABR for oligometastasis.

Name	Primary	No. of Mets	Treatments	Prior Treatment	Endpoint
SABR-COMET	NSCLC	≤5	SABR to all sites of disease vs. SOC	CT ≥ 4 weeks prior	OS
SARON-trial	NSCLC	≤3	SOC + conventional RT + SABR vs. Chemo	None	OS
STOMP	Prostate	≤3	Metastasis-directed therapy (surgery/SABR) vs. active surveillance	Surgery/RT or both	ADT-free survival
CORE	NSCLC, breast, prostate	≤3	SOC + SABR vs. SOC	CT ≥ 4–6 months prior	PFS
NRG BR002	Breast	≤2	SOC + SABR or surgery vs. SOC	≤6 months first-line CT	PFS, OS
NRG-LU002	NSCLC	≤3	Local consolidative therapy (SABR) + MT vs. MT alone	CT (at least 4 cycles)	PFS, OS

Mets: metastases, NSCLC: non-small cell lung cancer, SABR: stereotactic ablative radiotherapy, SOC: standard of care, CT: chemotherapy, RT: radiotherapy, OS: overall survival, PFS: progression-free survival, ADT: androgen deprivation therapy.

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
