# Peer review of "Stereotactic Radiotherapy for Oligometastasis"

_cancers, 2019, doi:10.3390/cancers11020133_

Reviewer 1 Report

Review

Stereotactic Radiotherapy for Oligometastasis

A:  Sotaro Otake, Taichiro Goto

 Overall

Review of work currently happening worldwide with SBRT for oligometastatic disease. Overall a bland summary which provides a useful review. Authors could suggest a future direction research could take

 Major

Please explain Table 1 – looks like all teh starting doses  are higher that the toxicity doses

Size of metastases is not discussed

Confusion between SRS ( intracranial ) and SBRT ( extracranial ) throughout paper. Need to follow current convention.

 Minor

Introduction

“Notwithstanding recent advances in pharmacotherapy, prognosis of metastatic cancer remains unsatisfactory.” Give at least one reference

38 medical doctors, mainly surgeons – simplify

42 widely various -poor English

43only a small number of studies. Give at least one reference

45 has emerged as a non-invasive alternative to surgery. Give at least one reference

48 presumption – you mean hypothesis

 89 the addition of SABR to whole brain radiotherapy – or addition of SRS? Intracranial? Not clear.

 121 - 2-3 localized metastases – what is a localised met?

 193 few overall treatment sessions - - you mean fractions?

 gamma knife was the prototype of SABR – or of SRS?

Author Response

Overall

Review of work currently happening worldwide with SBRT for oligometastatic disease. Overall a bland summary which provides a useful review. Authors could suggest a future direction research could take 

Response: The section “Future perspectives”was added to the manuscript, and we discussed the unanswered clinical questions of oligometastatic diseases in that section.

Major 

Please explain Table 1 – looks like all teh starting doses are higher that the toxicity doses

Response: In NRG-BR001 trial, if sufficient treatment related toxicity is observed in any of the metastatic locations at the initial starting dose, then that location will subsequently be treated with the dose limiting toxicity dose. If no treatment related toxicity is seen in any metastatic site, after treating sufficient patients, then the initial starting dose will be the recommended phase II dose. These explanations were added in the legend of Table 1.

 Size of metastases is not discussed 

Response:As to the size of oligometastasis, several studies identified large size as a poor prognostic factor and included it in their risk score systems. These descriptions were added to the manuscript.

 Confusion between SRS ( intracranial ) and SBRT ( extracranial ) throughout paper. Need to follow current convention. 

Response: The terms “SRS” and “SBRT” were rephrased into SABR as many as possible, because we think SABR is the most common term among them.

 Minor 

Introduction 

“Notwithstanding recent advances in pharmacotherapy, prognosis of metastatic cancer remains unsatisfactory.” Give at least one reference

Response: We added several references to this sentence.

 38 medical doctors, mainly surgeons – simplify 

Response: The phrase ‘ mainly surgeons’ was deleted.

 42 widely various -poor English 

Response: The phrase was corrected.

 43only a small number of studies. Give at least one reference

Response: We added several references to this sentence.

 45 has emerged as a non-invasive alternative to surgery. Give at least one reference

Response: We added several references to this sentence.

 48 presumption – you mean hypothesis

Response: We rephrased presumption into hypothesis, according to the reviewer’s suggestion.

 89 the addition of SABR to whole brain radiotherapy – or addition of SRS? Intracranial? Not clear. 

Response: In that clinical trial, SRS boost was added to whole brain radiotherapy. We revised the sentence, not to confuse the readers.

 121 - 2-3 localized metastases – what is a localised met?

Response: The term ‘localised’ was deleted.

 193 few overall treatment sessions - - you mean fractions? 

Response: This phrase means few fractions (shorter time period). Thus, we revised the phrase.

 gamma knife was the prototype of SABR – or of SRS?

Response: One of the first machines used to deliver SABR was the Gamma Knife, also termed as SRS, for intracranial tumors. We revised the sentence, not to confuse the readers.

Reviewer 2 Report

The authors contribute a review describing the role of stereotactic radiotherapy in oligometastasis. The main argument the authors attempt to convince the field is the definition of oligometastasis. Whether it is an intermediate stage between localized cancer and metastasized cancer. The author first established the biological basis for oligometastasis by presenting the current genomic and cell based experiments. Further discussion on the clinical aspects of oligometastasis invovles clinical significance and implication. The major focus of this review is the treatment of such disease and their pros and cons in clinical set-up. Using clinical data, the authors demonstrated the efficacy of SABR can achieve high rates of local control. 

Given the significance and novelty of this review, I believe it deserves publication in this journal after the authors may consider the following suggestions. 

Major: The major drawback of this paper is the lack of figures that summarize the biology and mechanism of oligometastasis. What are the differences between oligometastasis and other stages? The authors may consider using figures to vividly present the results. Otherwise, the readers may need some time to consume this review. As a review, figures are sometimes critical for high citations.

Minor, Please discuss the current diagnosis strategies of oligometastasis and compare them.

Author Response

Major: The major drawback of this paper is the lack of figures that summarize the biology and mechanism of oligometastasis. What are the differences between oligometastasis and other stages? The authors may consider using figures to vividly present the results. Otherwise, the readers may need some time to consume this review. As a review, figures are sometimes critical for high citations.

Response:We added 2 figures explaining the definition and mechanism of oligometastasis.

 Minor, Please discuss the current diagnosis strategies of oligometastasis and compare them.

Response: Multiple prognostic scores were developed for metastasectomy of oligometastatic disease, but Roberts et al. demonstrated that none of them could predict disease-free survival with sufficient discriminatory accuracy (Br J Surg, 2014; 101:856-66). Thus, limitations in accuracy of the risk scores have made it difficult to utilize them in clinical settings.

We added these descriptions to the manuscript.

Round  2

Reviewer 1 Report

Review of revision of:

Stereotactic Radiotherapy for Oligometastasis

A:  Sotaro Otake, Taichiro Goto

 Overall

Improved

Thanks for taking note of suggestions

 Major

Table one is still hard to understand.

 This statement:

“If sufficient treatment related toxicity is observed in any of the metastatic locations at the initial starting dose, then that location will subsequently be treated with the dose limiting toxicity dose. If no treatment related toxicity is seen in any metastatic site, after treating sufficient patients, then the initial starting dose will be the recommended phase II dose. “ is to me unethical. This means to me that you find a treatment related toxicity at a certain dose, then you treat more sites at that dose.  I would just delete the table and the detail of the study it refers to – just mention the study by name and move on. This is a trial that is not yet completed so removal of the detail wont impact the manuscript.

 Minor

Fig 2 – metasitasis  misspelt

Fig 2 – reddest bits are not metastasizing

 If these are addressed the manuscript does not have to come back to me for further review

Author Response

Comments and Suggestions for Authors

 Major 

Table one is still hard to understand. 

“If sufficient treatment related toxicity is observed in any of the metastatic locations at the initial starting dose, then that location will subsequently be treated with the dose limiting toxicity dose. If no treatment related toxicity is seen in any metastatic site, after treating sufficient patients, then the initial starting dose will be the recommended phase II dose. “ is to me unethical. This means to me that you find a treatment related toxicity at a certain dose, then you treat more sites at that dose.  I would just delete the table and the detail of the study it refers to – just mention the study by name and move on. This is a trial that is not yet completed so removal of the detail wont impact the manuscript.

Response: I deleted Table 1 and the detailed description regarding the trial, according to the reviewer’s suggestion.

Minor 

Fig 2 – metasitasis misspelt

Response: I corrected the misspelling in the figure. 

Fig 2 – reddest bits are not metastasizing 

Response:Metastasizing subclones may not have the highest metastatic potential. When cells with low metastatic potential incidentally adapt to the microenvironment and metastasize, they may induce oligometastasis.

We added these descriptions to the legend of Figure 2.

 Thank you very much for your thoughtful comments.